# Observation of secondary ice production in clouds at low temperatures.

Alexei Korolev[1], Paul DeMott[2], Ivan Heckman[1], Mengistu Wolde[3], Earle Williams[4], David J. Smalley[5] and Michael F. Donovan[5]

1. Environment and Climate Change Canada, Toronto, ON Canada
2. Colorado State University, Fort Collins, CO, USA
3. National Research Council Canada, Ottawa, ON, Canada,
4. Massachusetts Institute of Technology, Boston, MA, USA
5. MIT Lincoln Laboratory, Lexington, MA, USA

*Correspondence to*: Alexei Korolev ([alexei.korolev@ec.gc.ca](mailto:alexei.korolev@ec.gc.ca))

**Abstract**

Ice particles play an important role in precipitation formation and radiation balance. Therefore, an accurate description of ice initiation in the atmosphere is of great importance for weather prediction models and climate simulations. Despite the abundance of ice crystals in the atmosphere, the mechanisms for their formation remain not well understood. There are two major sets of mechanisms of ice initiation in the atmosphere: primary nucleation and secondary ice production. Secondary ice production occurs in the presence of preexisting ice, which results in an enhancement of the concentration of ice particles. Until recently, secondary ice production was mainly attributed to the rime-splintering mechanism, known as the Hallett-Mossop process, which is active in a relatively narrow temperature range from -3°C to -8°C. The existence of the Hallett-Mossop process was well supported by in-situ observations. The present study provides an explicit in-situ observation of secondary ice production at temperatures as low as -27°C, which is well outside the range of the Hallett-Mossop process. This observation expands our knowledge of the temperature range of initiation of secondary ice in clouds. The obtained results are intended to stimulate laboratory and theoretical studies to develop physically based parameterizations for weather prediction and climate models.

## 1. Introduction

Ice particles in the Earth's atmosphere play a crucial role in the modulation of precipitation and radiation transfer and eventually affect the hydrological cycle and climate on a global scale (e.g., Honget al., 2016; Matus and L'Ecuyer, 2017; Bacer et al. 2021). Despite their important role, a description of cloud processes involving ice particles is a subject of numerous challenges and uncertainties (Seinfeld, 2016). Understanding the mechanisms of ice initiation in the atmosphere is of a great importance for developing physically based parametrizations in weather prediction models and climate simulations (e.g., Muench and Lohmann, 2020).

There are two major mechanisms of ice formation in the atmosphere that are usually referred to as "primary" and "secondary". Primary ice production begins with the nucleation of ice particles either homogeneously in droplets supercooled below -38°C or heterogeneously on the surface of ice-nucleating particles (INP) through freezing of associated water or potentially directly from the vapor phase via deposition nucleation (e.g., Kanji et al. 2017). In contrast, secondary ice production (SIP) occurs in the presence of preexisting ice particles (e.g., Cantrell and Heymsfield, 2005; Field et al. 2017). Numerous observations have shown that the concentration of INPs in the atmosphere is generally lower than the concentration of cloud ice particles, and the difference between them may reach several orders of magnitude (e.g., Hobbs, 1969;

Mossop, 1985; Ladino et al. 2017). While the co-occurrence of both types of observations is still rare, the accumulated observations lead to the understanding that, in many cases, primary ice production cannot explain the concentrations of ice particles observed in clouds (Mossop, 1985; Cantrell and Heymsfield, 2005; Field et al. 2017). The excess of the ice particle concentration over that of INP was attributed to initiation of ice due to secondary ice production processes. At present, secondary ice production is recognized as one of

the major sources of ice particles in the atmosphere at temperatures above the temperature of homogeneous freezing, but with poor understanding as to the ways this comes about. It is worth noting that simulations of simple cloud situations do support closure of INPs and ice concentrations (Heymsfield et al., 1977; Eidhammer et al., 2010; Field et al., 2012.)

    There are six mechanisms identified as potential sources of SIP: (1) shattering during droplet freezing, (2)

the rime-splintering (Hallett–Mossop) process, (3) fragmentation due to ice–ice collision, (4) ice particle fragmentation due to thermal shock, (5) fragmentation of sublimating ice, and (6) the activation of ice-nucleating particles in transient supersaturation around freezing drops. A detailed review of these six SIP mechanisms is provided in Korolev and Leisner (2020).

    For many years, the rime splintering (Hallett-Mossop (HM)) mechanism (Hallett and Mossop, 1974;

Mossop and Hallett, 1974) was considered to be the main source of secondary ice in clouds. This perception of secondary ice initiation had been adopted by the cloud modeling community, and most of numerical cloud simulations described secondary ice production with the help of the HM-process only (e.g., ref. Morrison, 2005, Baser et al. 2021). Since the HM mechanism is active at relatively high temperatures ranging from -3C to -8C (Hallett and Mossop, 1974; Mossop and Hallett, 1974), secondary ice particles were activated in the

numerical cloud simulations in this temperature range only. Whereas, outside the HM-process temperature range ice initiation was assigned to primary ice nucleation only. Such approach may lead to underrepresentation of the role of secondary ice and result in biases in simulations (e.g., Qu et al, 2019, Huang, 2021).

    Recent laboratory studies (Lauber et al.,2018; Keinert et al., 2020) showed that droplet breakup during

freezing may contribute to formation of secondary ice at temperatures colder than the HM-process. Observations of glaciation of convective clouds also suggest that SIP may take place at temperatures colder than -8C (e.g., Lawson et al. 2015, 2017).

    The other four SIP mechanisms mentioned above may also contribute to ice formation outside the HM mechanism temperature range. In this regard, it is worth noting recent attempts to numerically explore the

effects of various SIP mechanisms across a wide temperature range (e.g., Phillips et al. 2017; Sullivan et al. 2018; Qu et al. 2019). However, parameterizations of SIP in cloud models are of debatable accuracy because the efficiencies of SIP mechanisms and the environmental conditions for required initiation of SIP are not understood at a fundamental level.

    In-situ observation of SIP is a challenging task. The most common way of identification of SIP is based on

comparisons of the observed concentration of ice particles and the concentration of INPs. Since in-situ airborne measurements of INP are not always possible, the INP concentration may be assessed from statistical dependence of INP concentration versus temperature (e.g., Kanji et al. 2017). Despite the fact that the INP concentration, at a specific temperature, may vary within over four orders of magnitude (e.g., Kanji et al. 2017), the observed concentration of ice particles frequently exceeds the maximum possible INP

concentration. Direct airborne in-situ observation of the SIP process is hindered by high aircraft speeds

(typically >100m/s), low sampling statistics of cloud particles, poor spatial coverage, and limited capability to perform Lagrangian measurements. In many cases, SIP particles may travel a long distance from the location of their origin to the location of their observation via sedimentation, turbulent diffusion or convective updrafts. Depending on their age, the secondary particles experience metamorphoses of shape and size due to varying ambient supersaturation S and temperature T and riming.  The concentration of SIP particles may also change due to the turbulent mixing, sedimentation, and aggregation. Therefore, in situ observation of secondary ice particles at the moment of their origin in many ways is a matter of luck, whether aircraft intersects the SIP cloud region at the right time and the right location.

There is a good wealth of the past and recent in-situ observations of SIP within the HM temperature range (e.g. Hallett et al., 1978; Crawford et al. 2012; Keppas et al. 2012; Lauber et al. 2021; Li et al. 2021; Luke et al., 2021; Ramelli et al. 2021 to name a few). However, there are fewer observations of SIP outside the HM temperature range (e.g. Hobbs, 1969; Costa et al. 2017, Lawson et al. 2017, 2022; Mignani et al. 2019; Pasquier et al. 2022). Most of these studies, reported observations of enhanced concentration of ice particles, which exceeded expected concentration of INPs at the temperature of observation. These enabled conclusions about their secondary production nature. However, location and environmental conditions associated with their origin and the age of the secondary ice particles mostly remained unknown.

This study presents an explicit observation of SIP in a strongly constraint cloud region at temperatures as low as -27C. This expands our knowledge of the temperature range of clouds where SIP may occur. The results of this study are important for understanding of one of the fundamental mechanisms of ice initiation in clouds. It is also expected that these observational results will stimulate further laboratory studies aimed at the exploration of SIP at low temperatures.

## 2. Results

The measurements were collected from the National Research Council Canada (NRCC) Convair-580 research aircraft. The NRCC Convair-580 was heavily instrumented for cloud microphysical measurements. The following instrumentation has been used in the frame of this study. Measurements of ice particle number concentration, ice water content (IWC), medium mass diameter (MMD) and maximum size of particles ($D_{max}$) were extracted from composite particle size distributions measured by imaging optical array probes (OAPs). These included a SPEC Inc. (Boulder, CO) two-dimensional stereo probe (2DS; Lawson et al., 2006), and a SPEC high volume precipitation probe (HVPS, Lawson et al. 1998). Cloud droplet size distributions were measured by both a PMS forward scattering spectrometer probe (FSSP; Knollenberg, 1981) and a DMT cloud droplet probe (CDP; Lance et al., 2010). High resolution particle images were measured with the SPEC cloud particle imager (CPI) (Lawson et al., 2001). A Rosemount icing detector was used for detection of liquid water at T < 4 °C (Mazin et al., 2001). Vertical velocity was measured by the Rosemount 858 (Williams and Marcotte, 2000) and Aventech AIMMS-20 (Beswick et al., 2008). Measurements of the air temperature were made with the Rosemount total-air temperature probes (model 102DJ1CG; Lawson and Cooper 1990; Friehe and Khelif, 1992). Dew and frost point temperatures were extracted from water vapor humidity measured by the Licor 7000 probe (LI-7000, 2007). The Convair-580 was also equipped with a NRCC airborne W-band radar (NAW) with Doppler capability (Wolde and Pazmany, 2005). The collected cloud microphysical data were processed and analyzed with the help of the ECCC D2G software.

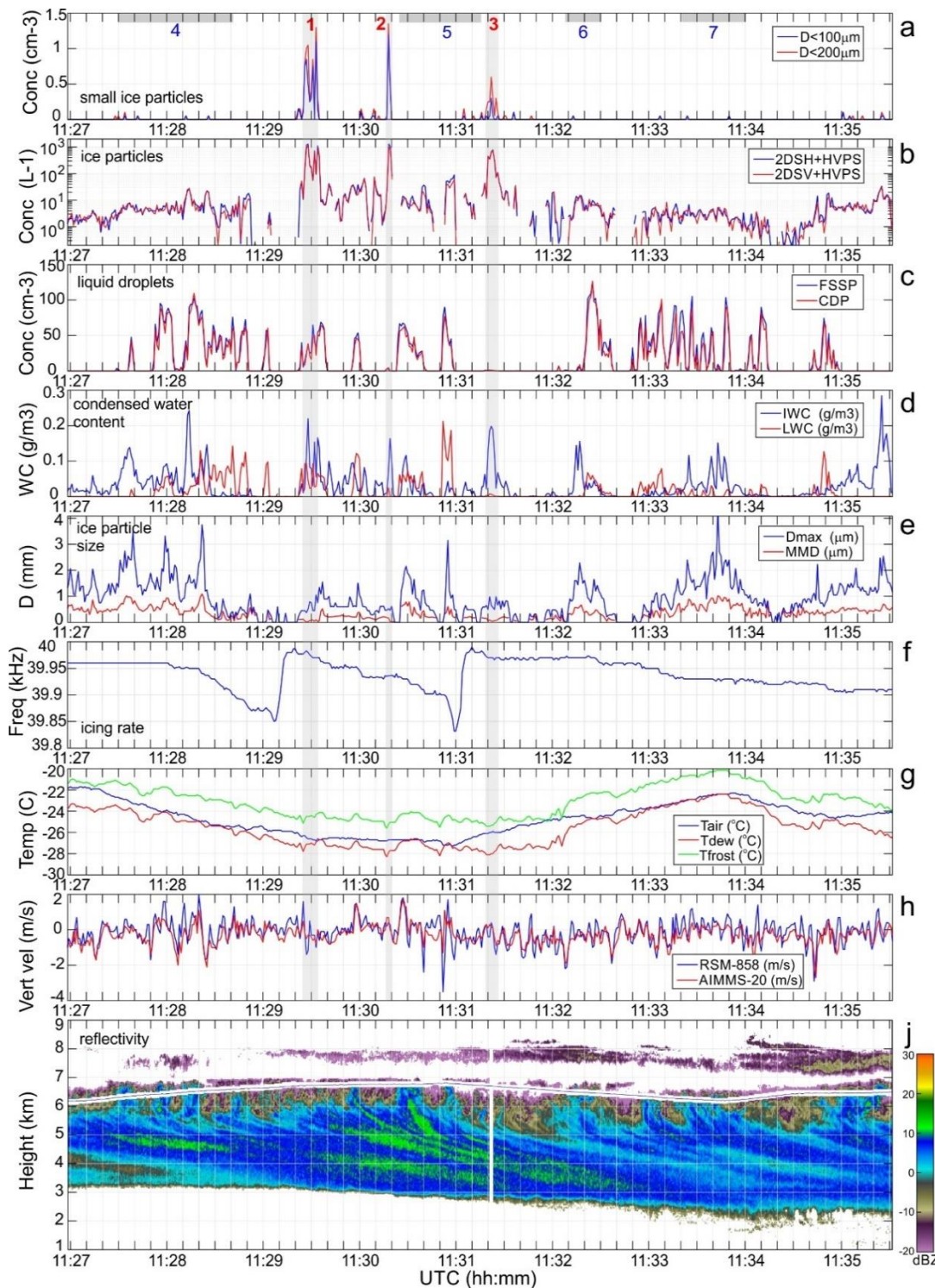

125

**Figure 1**. Time series of selected measurements (a) concentration of small pristine ice crystals with sizes smaller 100µm and 200µm assessed from CPI data; (b) concentration of ice particles >50µm measured by 2DS; (c) concentration of cloud droplets with 2µm<D<50µm measured by FSSP and CDP; (d) IWC and LWC calculated from 2DS+HVPS and FSSP measurements, respectively; (e) maximum ice particle size and median mass diameter of ice particles extracted from

130    2DS+HVPS data; (f) Rosemount Icing Cylinder frequency; (g) air, dew point and frost point temperatures measured by Licor-7000; (h) vertical wind velocity measured by RMS-858 and AIMMS-20; (j) reflectivity measured by W-band radar.

Figure 1 shows the time series of selected cloud microphysical and state parameters associated with the studied cloud segment. The data were collected during "porpoising" along the cloud top of the precipitating cirrocumulus-nimbostratus (Cc-Ns) cloud system (Fig.1j). The Cc-Ns was overlaid by another thin cirrostratus (Cs) layer with the cloud top at approximately 8km, which was separated from the lower Cc-Ns by a few hundred meters of a cloud free layer. The morphology of the cloud top can be seen from the GOES-16 satellite visible and infrared images in Fig. A1 (supplementary material).

The aircraft altitude during the porpoising changed between 6200m and 6800m (Fig.1j) and the temperature varied from -22C to -27C (Fig.1g). From a microphysical standpoint, the environment in the studied cloud was highly inhomogeneous, consisting of intermittent mixed-phase and ice cloud segments. The presence of supercooled liquid water is verified by the changing frequency of vibrating icing cylinder (Fig.1f), when passing through the liquid-containing cloud regions (Fig.1c,d). The horizontal extension of mixed-phase cloud regions varied from a few hundred meters to a few kilometers (Fig.1c,d), with liquid water content (LWC) peaking up to 0.2g/m$^3$. The average concentration of liquid droplets the mixed phase clouds was from 46cm$^{-3}$ peaking up to 120cm$^{-3}$ (Fig.1c), and the mean volume diameter (MVD) changing between 8$\mu$m and 15$\mu$m. The probability density function and size distributions of cloud droplet concentration and LWC are shown in Figs. S2.

The high variability in the cloud microstructure was likely induced by an intense turbulence. The vertical velocity varied from -2m/s to +2m/s with $\sigma$=0.6m/s (Fig.1h). Vertical velocity $U_z$>0.1 to 0.5m/s is sufficient to activate liquid water in preexisting ice clouds (Korolev and Mazin, 2003) and maintain a mixed-phase environment (Hill et al. 2013; Field et al. 2013). The interaction between ice particles and newly formed liquid droplets will occur through riming and Wegener-Bergeron-Findeisen processes (Wegener, 1911; Bergeron, 1935; Findeisen, 1938), which may result in a complete depletion of liquid water by ice particles and glaciation of the mixed-phase cloud. Intense turbulence may also stimulate entrainment of the dry air through the cloud top. This will result in the evaporation of cloud droplets and ice particles, which contributes to further increases in cloud inhomogeneity and expedites glaciation.

Figure 1b shows the time series of cloud particles concentration with a maximum size $D_{max}$>40$\mu$m, which was calculated from a composite particle size distribution measured by the 2DS and HVPS. The 2DS binary imagery does not allow segregation of the phase state of small ice particles ($D_{max}$<80$\mu$m) because of poor pixel resolution (Korolev et al. 2017). However, analysis of the high-resolution CPI imagery (2.3$\mu$m) suggests that no droplets with $D_{max}$>40$\mu$m were present in these cloud regions, and therefore, particles $D_{max}$>40$\mu$m with a high level of confidence can be considered as ice.

The most striking observation in the studied cloud is three cloud segments indicated by numbers 1-3 in Fig.1a with the concentration of ice particles varying in the range of 200 < $N_{ice}$ < 1200 L$^{-1}$ (Fig.1a,b). However, elsewhere around these cloud segments, the background concentration of ice particles varied from 0.4L$^{-1}$ to 30L$^{-1}$ at the levels 5 and 95 percentiles, respectively, with the mean value 7.5L$^{-1}$ (Fig.S3). There is nearly 2-3 orders magnitude of difference between the background and enhanced ice concentrations and simultaneous measurements of high ice concentrations by two independent instruments (Fig.1a,b) exclude explanation of this observation by statistical fluctuations of particle counts.

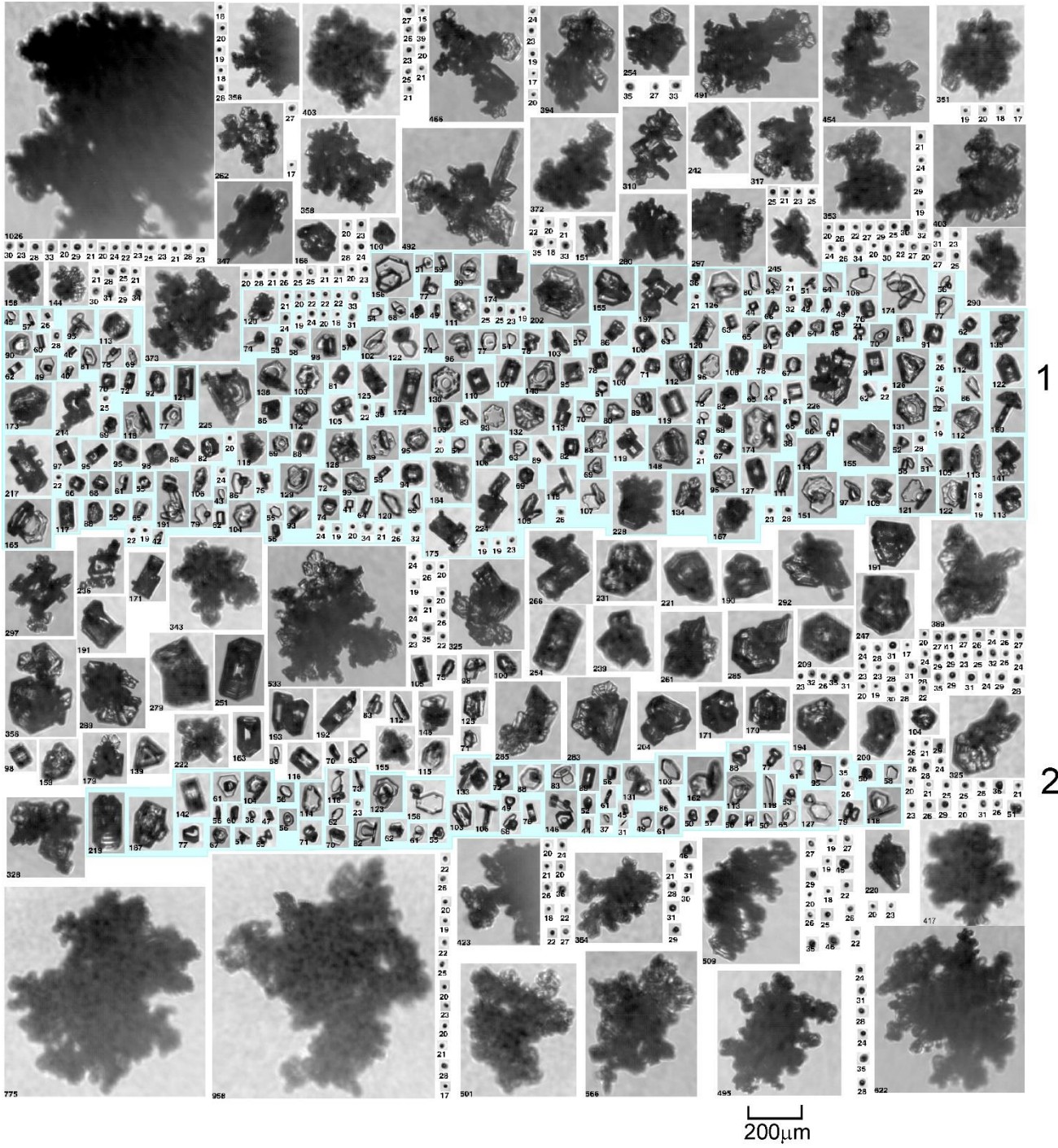

170

**Figure 2**. Images of cloud particles sampled by CPI during traverse of a cloud shown in Fig.1. First image 11:28:22 UTC, last image 11:30:34 UTC. The numbers at the left bottom corner of each image indicate the maximum image size in μm. The images associated with the high ice concentration cloud regions 1 and 2 in Fig. 1a appear on a blue background.

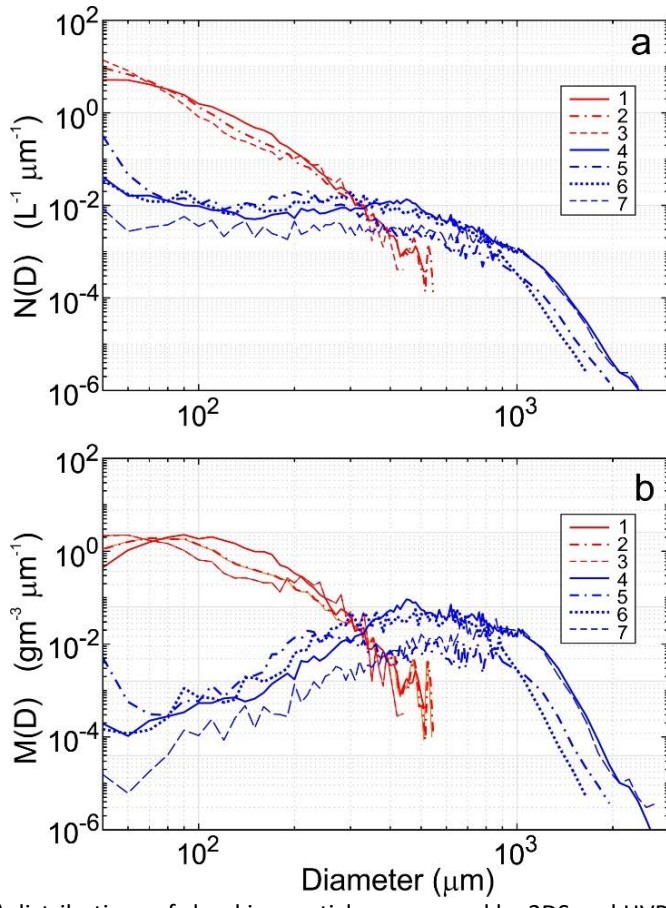

**Figure 3**. Size (a) and mass (b) distributions of cloud ice particles measured by 2DS and HVPS in cloud regions in Fig.1 indicated by numbers 1-7. Size and mass distributions 1-3 (red) correspond to the cloud regions with high concentration of small ice particles; 4-7 (blue) correspond to the cloud regions with aged ice.

Figure 2 shows a sequence of the high-resolution CPI images measured during a traverse through the cloud region with segments 1 and 2 (Fig. 1a, 11:28:22 - 11:30:34 UTC). As shown in Fig.2, the particles inside the regions of enhanced ice concentration are mostly small facetted hexagonal plates and columns, whereas outside regions 1-3, the ice particles have irregular shape and many of them are covered by fresh or aged rime.

Figure 3 presents average composite size and mass distributions measured by 2DS and HVPS probes in seven cloud segments shown in Fig.1a. Three of these segments are associated with the cloud regions with enhanced concentrations (1-3, Fig.1a) and the other four are associated with the neighboring regions 4-7, indicated by grey strips in Fig.1a. In Fig.3, the distributions in the cloud segments with high concentration (1-3 red) are grouped close to each other, and they are quite different from the distributions (4-7, blue) in the neighboring cloud regions. The maximum particle size $D_{max}$ in cloud segments 1-3 is limited to a range 400-600mm, whereas in the background cloud segments 4-7 the $D_{max}$ values reach 1.5mm to 2mm. The time series of $D_{max}$ and mean mass diameter (MMD) are also shown in Fig. 1e.

The obtained observations suggest that the formation of a high concentration of small ice particles in cloud regions 1-3 can be attributed to a physical process rather than to the statistics of sampling. A valid question arises: What is the mechanism responsible for the formation of the high concentration regions?

Based on INP in-situ measurements, the maximum concentration of primary ice particles at T=-27C may vary from approximately $10^{-1}L^{-1}$ to $1000L^{-1}$ (e.g., Kanji et al. 2017, Petters and Wright, 2015). Therefore, the observed concentration $N_{ice}$=1200 $L^{-1}$ might be explained by primary ice nucleation. On the other hand, the background concentration of ice particles in the neighboring cloud regions is systematically lower by 1-2 orders of magnitude than $N_{ice}$ in the cloud segments 1-3 (Fig.1b). It would be reasonable to assume that the primary ice particles were initiated by the same population of INPs, giving a concentration of ice varying between $0.4L^{-1}$ to $30L^{-1}$. The rapid increase of the concentration of INPs by 1-2 orders of magnitude in a spatially limited area is an unlikely explanation. Such spatial inhomogeneities of the INP concentration would be rapidly mixed with the surrounding environment due to turbulent diffusion. Assessment of the turbulent energy dissipation rate ($\varepsilon$) from Fig.1h and the maximal horizontal extension (L) of the cloud segments 1-3 from Fig.1a yields $\varepsilon \approx 10^{-2} m^2/s^3$ and $L \approx 10^3 m$, respectively. Therefore, the mixing time could be assessed as $\tau_m = (L^2/\varepsilon)^{\frac{1}{3}} \sim$ 5x$10^2$s. Such a mixing time is much shorter than the age of the existing Cc-Ns cloud layer from the GOES-16 satellite imagery as at least 1h. At time scales $\tau > \tau_m$ the spatial variations of the INPs will be homogenized due to mixing with the ambient environment. Therefore, the explanation of the enhanced concentration of ice particles due to spatial inhomogeneity of the INP concentration can be ruled out.

Another possibility explaining the enhanced ice concentration may be related to the droplet freezing. The rate of droplet freezing has been assessed here with the help of the Bigg's equation (Bigg, 1953; Khain et al. 2021). For the droplet size distribution averaged over the cloud span (Fig.S2a) it was found that at -27°C the rate of droplet freezing is approximately $dN_{ice}/dt \approx$0.3 $L^{-1}s^{-1}$ (see supplementary material). Therefore, in order to reach an enhance ice concentration of the order of $10^3L^{-1}$ the residence time of the cloud parcel should be $N_{ice}/dN_{ice}/dt \approx$ 0.92h. This is an unrealistically long residence time for a cloud parcel in a stratirom cloud layer of a few hundred meter depth. During this time the turbulent diffusion will smear the entire cloud parcel as well as ice particles mitigating formation of sharp gradients of ice concentration as in Fig.1a,b. All these, makes the "droplet freezing" hypothesis insufficient to explain the observed enhanced concentration of ice. The enhanced concentration of ice can possibly be explained by seeding from the cirrus cloud overlaying the Cc-Ns layer (Fig.1j). However, the W-band radar measurements indicated that the two cloud layers were separated by approximately 500 meters with no radar return (Fig.1j; 11:29 - 11:32 UTC). On the other hand, measurements of humidity during occasional climbing above the cloud top of the Cc-Ns layer (not in Fig.1) showed that the two cloud layers were separated by dry air. The dry layer will hinder seeding due to sublimation of ice particles. A few random ice particles, which may survive sublimation in the dry layer and can reach the Cc-Ns layer are unlikely to explain the high concentration of ice in segments 1-3. Therefore, seeding from the overlayed cirrus cloud also does not seem to be a feasible explanation of high ice concentration.

Secondary ice production appears to be the most plausible reason of the enhanced concentration of ice in cloud segments 1-3. This explanation is supported by the numerous small pristine ice particles in these cloud regions (Fig.2). Very similar small pristine ice crystals were observed in studies of Korolev et al (2020), Lauber et al. (2021) at subfreezing temperatures.

The size of individual facetted ice crystals in the enhanced ice concentration cloud segments 1-3 with enhanced ice concentration varied from 26μm to approximately 170μm (segment 1, Fig.2), from 31μm to approximately 142μm (segment 2, Fig.2), and from 61μm to approximately 250μm (segment 3, not shown). Ice particles with larger sizes are either polycrystalline, aggregates or rimed. The size span between smallest

and largest crystals indicates that the SIP occurred not instantly, but rather was extended over some time. Assuming the initial size of secondary ice particle is 5μm (Korolev et al. 2020) and the humidity is saturated over liquid water the time required to grow ice particles to the maximum size indicated above can be estimated as approximately 160 s (segment 1), 115s (segment 2), and 360s (segment 3).

In reality, the in-cloud humidity is continuously changing because of mixing with the neighboring environment and on average it has a tendency to decrease due to depletion of water vapor by ice particles. Therefore, the above assessment yields a lower estimate of the ice crystals growth time. The actual growth time will be longer given the lower RH compared to its saturated-over-water value.

Figure 1g shows a time series of the frost point ($T_f$), dew point ($T_d$), and air temperature ($T_a$). These temperatures enable assessment of relative humidity over ice $RH_{ice}$. As seen from Fig.1g in cloud regions with high ice concentrations the cloud environment was always supersaturated with respect to ice (i.e. $T_f>T_a$), and $RH_{ice}$ varied in the ranges 112% < $RH_{ice}$ < 130% (segment 1), 113% < $RH_{ice}$ < 119% (segment 2), 107% < $RH_{ice}$ < 111% (segment 3). Saturation over water was reached in segment 1 (i.e., when $T_d≈T_a$), whereas segments 2 and 3 were undersaturated with respect to water.

Supercooled liquid droplets might have been initially present in segments 1-3 before the SIP process had begun. However, the initiation of a large amount of secondary ice would intensify the WBF process and expedite glaciation of the mixed-phase environment. Assuming an initial LWC=0.1g/m³ and a concentration of ice particles $N_{ice}$=500-1000L⁻¹, the assessment of the glaciation time (Korolev and Mazin, 2003) yields $\tau_{gl}$=60-90s.

This obtained assessment of the glaciation time and growth time of ice crystals allows for an estimate of the age of the SIP cloud segments 1-3, which is approximately 2-5min. The following growth of ice particles will result in their sedimentation and formation of virgae, which are quite noticeable in the W-band radar returns in Fig. 1j. Luke et al. (2021) observed similar virgae in Arctic stratiform clouds in regions associated with SIP.

At that stage it does not seem feasible to identify which SIP mechanism is responsible for the observed enhancement of ice concentration. Observation of heavily rimed particles suggests that the rime-splintering mechanisms might be active. Unfortunately, early experimental studies of rime-splintering were mainly focused on relatively high temperatures (e.g., Aufdermaur and Johnson, 1972; Hallett and Mossop, 1974; Mossop, 1976; Heymsfield and Mossop, 1984; Saunders and Hosseini, 2001) and there were no published results on efficiency of rime-splintering at temperatures lower that -18C (Latham and Mason, 1961). Droplet breakup during freezing is another plausible SIP mechanism to explain the observations (Lauber et al.,2018; Keinert et al., 2020; Staroselsky et al., 2021). It is worth mentioning that the droplet breakup during freezing and rime-splintering is supported by the presence of liquid phase in this layer. Absence of supercooled liquid in segments 2 and 3 may be explained by glaciation of the mixed-phase environment. Developed shapes of rimed ice particles (Fig.2) with a large number of seemingly fragile branches suggests the ice-ice collisional breakup mechanism is another plausible candidate for explaining the enhanced concentration of ice (Vardiman, 1978; Takahashi et al. 1995). Shattering of fragile ice branches resulting from a thermal shock during freezing (e.g., King and Fletcher, 1976) and ice nucleation in high-supersaturated wakes behind riming ice particles (e.g. Gagin, 1972; Prabhakaran et al., 2020) also cannot be ruled out. However, fragmentation during ice sublimation (Oraltay and Hallet, 1989, Bacon et al., 1998) appears to be the least plausible mechanism, since no undersaturated environment was observed in the studied cloud layer (Fig.1g).

As follows from the above, no clear preferences could be granted to any of the five potential SIP mechanisms. However, in absence of credible experimental data on efficiency and environmental conditions required for each SIP mechanism, the above discussion on the feasibility of SIP mechanism bears a speculative character. It is worth mentioning that an unknown mechanism responsible for the observed enhanced concentration of ice also cannot be ruled out.

It is interesting to note that, in the stratiform layers, SIP occurred in spatially localized cells where the necessary and sufficient conditions for SIP initiation were met. The horizontal extension of the SIP regions in Fig.1a is estimated to vary from approximately 500m to 1km.

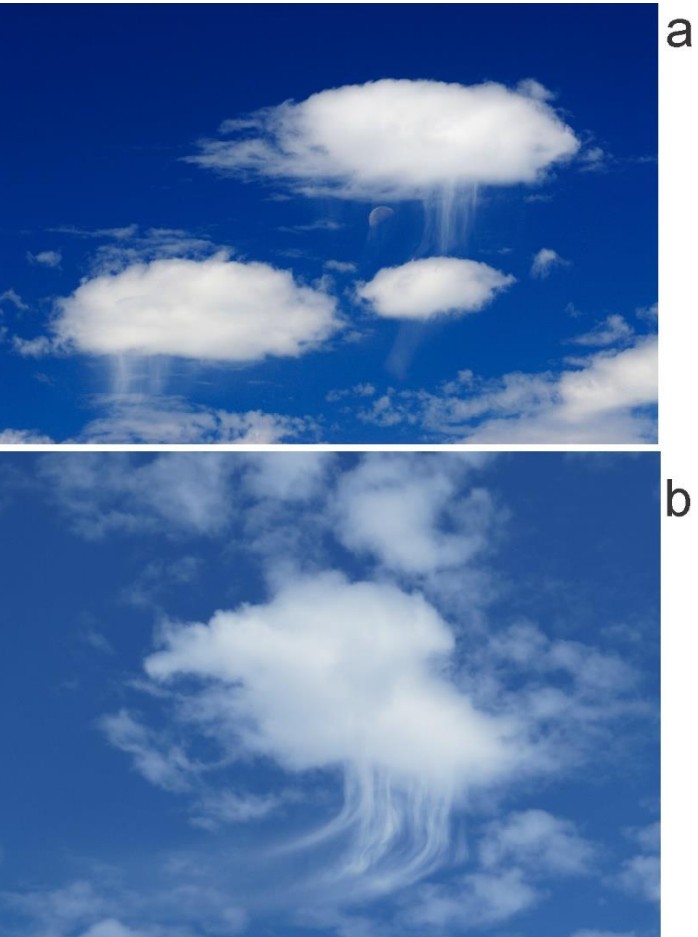

**Figure 4**. Images of clouds with ice virga falling out of clouds. Optically dense cores of the clouds sourcing the virga indicate on presence of liquid droplets. Origin of the figures: (a) courtesy: Kaufung/ Alamy Stock Photo/ CRDP4A; (b) courtesy Dr. Martin Gudd (Institute for Professional Weather Education, https://www.weather-education.de )

The obtained results can be illustrated by pictures of altocumulus and altostratus clouds with virgae. The optical density of the main bodies of the clouds indicates that these clouds are dominated by liquid droplets. Ice clouds usually have lower optical density, and they are more transparent given the lower concentration of ice particles compared to that of liquid droplets. The streaky structure of the virgae with relatively small vertical extension of the clouds in Fig.4 indicates that the particles precipitating out of the clouds are ice. Usually, formation of liquid precipitation requires deep liquid layers compared to those in Fig.4. A specific point of the photos in Fig.4 is that virgae of ice particles did not extend across the entire cloud, but rather formed in very local regions. Such formation of is unlikely to be explained by primary nucleation due to spatial

fluctuations of INPs, which formed a region with an enhanced concentration of INPs. The most plausible explanation is that the virgae in Fig.4 are a result of SIP at the locations where the relevant SIP conditions were satisfied.

### 3. Conclusions

This is a first explicit in-situ observation of SIP at temperatures down to -27C. This expands our understanding on the temperature range, where SIP may occur in natural clouds. Even though laboratory studies suggest that SIP may take place at temperatures colder than that relevant to the HM process, there were no unambiguous observations of SIP in natural clouds at temperatures as low as -27C. The obtained results are important to stimulate laboratory and theoretical studies to identify SIP mechanisms at low temperatures. One of the key objectives along this way is finding of necessary and sufficient conditions for SIP. This would facilitate development of physically based parameterizations for NWPs and climate models.

*Data availability:* The cloud microphysical datasets are available on Zenodo (https://doi.org/10.5281/zenodo.7075925).

*Author contribution*: AK performed conceptualization, data collection and data analysis and drafting the paper, PD interpretation of INP and ice particle concentration, IH data processing, MW organizing the Convair-580 flight operation, collection and processing radar data, EW, DS, MD arranging BAIRS-2 field campaign, writing, reviewing, and editing the manuscript.

*Competing interests*: The authors declare that they have no conflict of interest.

*Acknowledgements*: Many thanks to NRCC and ECCC tech personal for the integration of the airborne instrumentation onboard the NRCC Convair-580, probes' maintenance, and data collection. Special thanks to the NRCC pilots Anthony Brown and Robert Erdos for operation of the NRCC Convair-580 during the data collection. This work was supported by ECCC, NRCC, MIT and FAA.

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
