# Peer review of "Observation of secondary ice production in clouds at low temperatures."

_EGUsphere, 2022_

## Community Comment (CC1)

Review of the study "**Observation of secondary ice production in clouds at low temperatures**", authored by

Alexei Korolev, Paul DeMott, Ivan Heckman, Mengistu Wolde, Earle Williams, David J. Smalley and Michael F. Donovan.

The study provides the first in-situ observation of secondary ice production at temperatures as low as -27°C. These observations are unique and important. I recommend to accept the paper with minor revisions.

Minor comments are:

1. line 40. The formation of ice by droplet freezing is not mentioned. Since the temperature measured was higher than the temperature of homogeneous drop freezing, the standard immersion drop freezing takes place at T=-27C is. Can the authors evaluate the rate of this immersion freezing of drops (for instance using standard Bigg formula)? Note that according to the observations presented in this study, the zones of high concentration of ice crystals coincide with the zones of significant peaks in droplet concentration and LWC.

   Please discuss the possible role of the freezing of drops, which concentration is several orders higher than that of ice crystals, in production of cloud ice. It would be reasonable to refer in this context the study by Khain et al. (2022), in which A. Korolev is a co-author.

   Can you compare the rates of immersion drop freezing and the rates of the mechanisms of primary ice nucleation mentioned in the paper?

2. Line 68. The study Qu et al, (2019) is not presented in the reference list. If you mean the study published in J. Geophys. Res. (see list of references below), that study shows that only secondary ice production can explain in-situ observations. The fraction of ice produced by primary nucleation was evaluated as several per cents at all temperatures. In my view, the study by Qu et al, (2019) was the first paper that reproduced size distributions ice and water observed in-situ measurements and showed that SIP plays a crucial role in the formation of such distributions. So, the statement in the current study that "Such approach may lead to underrepresentation of the role of secondary ice and result in biases in simulations" is not attributed to the study by Qu et al. (2019).

3. Line 75. Here reference to Qu et al. (2019) as well as to Phillips et al. (2017) should be included. In these studies simulations with a bin-microphysics cloud model reproduced ice size distributions formed by SIP by drop-ice and ice-ice collisions.

4. Line 78. The important attempt to understand the fundamental mechanisms of SIP by drop freezing was carried out by Staroselsky et al., 2021.

5. Line 128. Fig. 1. Please pay attention on the high correlation between droplet concentration and LWC, on the one hand, and the concentration of ice particles. In my opinion, this correlation shows the key role of drops in the formation of ice particle concentration. I believe that this high correlation decreases the number of possible SIP mechanisms, at least in the present case study.

References

Khain A, Pinsky M., and A. Korolev, 2022: Combined effect of the Weber-Bergeron-Findeisen mechanism and large eddies on microphysics of mixed-phase stratiform clouds. J. Atmos. Sci, Volume 79: Issue 2, 383–407, https://doi.org/10.1175/JAS-D-20-0269.1

Phillips V., J-I. Yano, M. Formenton, E. Ilotoviz, V. Kanawade, I. Kudzotsa, J. Sun, A. Bansemer, A. Detwiler, A.P. Khain and S. Tessendorf, 2017: Ice multiplication by break-up in ice-ice collisions. Part 2: Numerical simulations. *J. Atmos. Sci.*, 74, 2789 – 2811.

 Staroselsky A., R.Acharya, and A. Khain, 2021: Toward a theory of the evolution of drop morphology and splintering by freezing. J. Atmos. Sci. 78, 10, 3181–3204, https://doi.org/10.1175/JAS-D-20-0029.1

Yi Qu, A. Khain, Vaughan Phillips, Eyal Ilotoviz, Jacob Shpund, Sachin Patade, Baojun Chen, 2019: The role of ice splintering on microphysics of deep convective clouds forming under different aerosol conditions: simulations using the model with spectral bin microphysics. *J. Geophys. Res*. 125, Issue3;,125, e2019JD031312.https://doi.org/10.1029/2019JD031312

---

## Author Response (AR1)

**Response to Reviewer 1 (Thomas Leisner) of "Observation of secondary ice production in clouds at low temperatures" authored by A. Korolev et al. (egusphere-2022-491)**

The authors thank the reviewer and the editor for the positive evaluation of the paper and their constructive comments. In our response below, the reviewer's comments are in black, and our responses are in blue.

The authors report in situ aircraft observations of sharply constrained regions within a nimbostratus cloud with a strongly enhanced ice number concentration due to small and pristine ice particles.

They attribute this to secondary ice processes being effective at the low (~ -26°C) temperatures of the observation.

The observations have been made with a very comprehensively equipped aircraft in an important type of cloud and the presented data are of very high quality. Therefore, the dataset is of substantial scientific interest and should be published. I will most certainly stimulate discussion and advance secondary ice understanding.

I have one main point with the discussion and interpretation of the data that should be addressed before publication and a few minor remarks.

**Main point**:

1.   The authors exclude primary ice nucleation because turbulent diffusion should smear out INP concentrations sufficiently to exclude very localized maxima. On the other hand, they invoke turbulence as the main cause of the observed strong heterogeneity in this cloud. Might this also be used to explain the observations with primary ice nucleation? E.g. a layer of dust-rich air might be entrained from cloud top?

**Reply**: We feel that such an occurrence is highly unlikely, due to the location, timing of and strength of dust transport events reaching Eastern Canada, and due to the concentration of INPs that are likely to be present in such dust layers. First, it is certainly true that long range dust layers can occur by late March over North America, but they are episodic and infrequent events at a scale needed to have the noted impacts. By way of example, we can consider a weaker and stronger dust loading at cloud top elevation and flex a few parameterizations for INPs in these cases. To frame a weaker versus stronger event, we must allude to the literature, as documented observations of such over Eastern Canada are rare. We will pick three studies for this purpose, though many would suffice to provide similar constraint. First, we reference Shi et al. (2022) (https://doi.org/10.5194/acp-22-2909-2022), a study of lower and higher latitude dust reaching the Arctic. For lower latitude dust, the same paper shows that there is not much difference between what reaches the Arctic circle and most of Canada. In March, this paper shows that dust mass concentrations may range from a low of about 0.1 to a high of 10 $\mu g \, m^{-3}$. Comparing to the study of Richardson et al. (2009; doi:10.1029/2006JD007500) of Asian dust transports over the

Western U.S., we see a similar range of values reflected at different times at the surface and aloft. Looking at Zhang et al. (2019; https://doi.org/10.1029/2019JD031388) for studies over the NE United States, closer to the site, a typical springtime maximum is 1 $\mu$g m$^{-3}$. Thus, we will use mass loadings of 1 and 10 $\mu$g m$^{-3}$ to frame a typical dust transport event. We may translate these into surface area concentrations of roughly 0.5 and 5 $\mu$m$^2$ cm$^{-3}$, although the explicit conversion depends on the dust size distribution. We think that these are reasonable estimates for regions away from close proximity to major dust source regions and their immediate transport paths. Zhang et al. (2019) also includes estimates of aerosol concentrations at sizes larger than 0.5 $\mu$m, and this allows us to also assume a range of such concentrations to align with our mass and surface area concentrations, and thereby flex two different mineral dust INP parameterizations. Peak values at 4 km in spring are 0.5 cm$^{-3}$ for events in the upper 10% predicted by the GEOS-Chem model in that study. We will use that as our "weak" case and we will use 5 cm$^{-3}$ as an extreme, not even observed in their data set. We use Niemand et al. (2012) for surface site density approach and DeMott et al. (2015) for an aerosol number concentration approach.

At -25°C, we find:

| Parameterization | Weak case | Strong case |
|---|---|---|
| Niemand | 3.7 L$^{-1}$ | 37 L$^{-1}$ |
| DeMott | 1.1 L$^{-1}$ | 20.3 L$^{-1}$ |

We may note that these concentration ranges match quite well with the range of INP concentrations we report. However, we may also inspect global aerosol model output for the time frame of the cloud measurements in this study. Referencing the Naval Research Laboratory NAAPS model, https://www.nrlmry.navy.mil/aerosol_temp/loop_html/aer_globaer_noramer_loop_2017032600. html#, we find that although there was some dust generation within the S. Central United States at the time of flights, the general weather patterns placed this far from any chance to impact SE Canada. As well, there appear no surface influences of long-range transport, for example from Asia across N. America at the time. These are surface plots but as we said will usually somewhere reflect at the surface values comparable to what is aloft, and no other satellite or model products available at the NRL model site or elsewhere show any evidence of a dust plume event anywhere in Canada at the time of these cloud measurements. This supports that no event more extreme than we have assumed could have been present at the time of the study.

DeMott, P. J., A. J. Prenni, G. R. McMeeking, Y. Tobo, R. C. Sullivan, M. D. Petters, M. Niemand, O. Möhler, and S. M. Kreidenweis, 2015: Integrating laboratory and field data to quantify the immersion freezing ice nucleation activity of mineral dust particles, *Atmos. Chem. Phys.,* **15**, 393–409.

Niemand, M., O. Moehler, B. Vogel, H. Vogel, C. Hoose, P. Connolly, H. Klein, H. Bingemer, P. DeMott, J. Skrotzki, and T. Leisner, 2012: Parameterization of immersion freezing on mineral dust particles: An application in a regional scale model. *J. Atmos. Sci*, **69**, 3077-3092.

Richardson, M. S., P. J. DeMott, S. M. Kreidenweis, D. J. Cziczo, E. Dunlea, J. L. Jimenez, D. S. Thompson, L. L. Ashbaugh, R. D. Borys, D. S. Westphal, G. S. Cassucio and T. L. Lersch, 2007: Measurements of heterogeneous ice nuclei in the Western U.S. in springtime and their relation to aerosol characteristics. *J. Geophys. Res.*, **112**, D02209, doi:10.1029/2006JD007500.

Shi, Y., Liu, X., Wu, M., Zhao, X., Ke, Z., and Brown, H.: Relative importance of high-latitude local and long-range-transported dust for Arctic ice-nucleating particles and impacts on Arctic mixed-phase clouds, Atmos. Chem. Phys., 22, 2909–2935, https://doi.org/10.5194/acp-22-2909-2022, 2022.

Zhang, Y., Luo, G., & Yu, F. (2019). Seasonal variations and long-term trend of dust particle number concentration over the northeastern United States. Journal of Geophysical Research: Atmospheres, 124, 13,140–13,155. https://doi.org/10.1029/ 2019JD031388

2.  Vice versa one could argue that turbulence should smear out the conditions favorable for SIP as well.

**Reply**: Turbulence in stratiform cloud layers, such as Sc, Ac, and especially near their cloud tops, is frequently anisotropic (e.g., Pedersen et al. 2018; Lin and Pao, 1979). The anisotropic turbulent motions feedback microphysical response. One of the manifestations of the cloud anisotropy is a well documented cellular structure of precipitation, which is observed on radar imagery. This can also be seen in the radar returns shown in Fig.1j. Therefore, the authors consider that turbulence is stratiform clouds could facilitate formation of conditions favorable for SIP initiation, e.g., through generation enhancing riming or collision coalescence process with the following formation of fragile rimed ice particles and drizzle size drops, respectively, rather than smear out SIP conditions. Fragile rimed ice particles and drizzle size drops would facilitate ice collisional breakup and fragmentation during freezing SIP mechanisms, respectively.

Lin, J.T. and Pao, Y.H. 1979: Wakes in stratified fluids. Annu. Rev. Fluid Mech. 11, 317–338.

Pedersen, J. G., Ma, Y.-F., Grabowski, W. W., and Malinowski, S. P. 2018: Anisotropy of observed and simulated turbulence in marine stratocumulus. Journal of Advances in Modeling Earth Systems, 10, 500–515. https://doi.org/10.1002/2017MS001140

3.  It seems that all 6 mechanisms discussed in Korolev and Leisner (2020) could be dismissed due to the uniformity and scarcity of larger ice particles as seen from Fig. 3 (presence of larger ice particles would favor mechanisms 2, 3, 6) and larger liquid drops (which would favor mechanisms 1, 4) and due to the fact that temperatures were continuously below the frost point (excludes mechanism 5).

Therefore, any attribution of the observed small ice to either primary nucleation or SIP seems hard to justify. I suggest that the authors discuss these issues somewhat more in detail.

**Reply**: The authors agree that at that stage, the identification of a potential SIP mechanism responsible for the observed enhancement of the ice concentration is not plausible. To address this concern, the authors made a disclaimer: "*As follows from the above, no clear preferences could be granted to any of the five potential SIP mechanisms. However, in absence of credible experimental data on efficiency and environmental conditions required for each SIP mechanism, the above discussion on the feasibility of SIP mechanism bears a speculative character*." To strengthen the above disclaimer the following statement was added: "*It is worth mentioning that an unknown mechanism responsible for the observed enhanced concentration of ice also cannot be ruled out.*"

4.  Nevertheless, I agree with the authors that some form (maybe even a hitherto not invoked process) of SIP is "the most plausible reason" (line 216)…

**Reply**: Thanks for the comment.

5.  …but I find the statement in the abstract "the first in-situ observation of SIP at temperatures as low as -27°C" too strong.

**Reply:** The aforementioned statement was replaced by the following: "*The present study provides an explicit in-situ observation of secondary ice production at temperatures as low as -27°C, …*"

We also added a paragraph to clarify the objectives of these paper with references to the in-situ observations of SIP at temperatures below the HM temperature range:

"There is a good wealth of the past and recent in-situ observations of SIP within the HM temperature range (e.g. Hallett et al., 1978; Crawford et al. 2012; Keppas et al. 2012; Lauber et al. 2021; Li et al. 2021; Luke et al., 2021; Ramelli et al. 2021 to name a few). However, there are fewer observations of SIP outside the HM temperature range (e.g. Hobbs, 1969; Costa et al. 2017, Lawson et al. 2017, 2022; Mignani et al. 2019; Pasquier et al. 2022). Most of these studies, reported observations of enhanced concentration of ice particles, which exceeded expected concentration of INPs at the temperature of observation. These enabled conclusions about their secondary production nature. However, location and environmental conditions associated with their origin and the age of the secondary ice particles mostly remained unknown."

**Minor remarks**:

6.  It is hard to assign the ice particle images in Fig. 2 to Fig. 1 as the attribution to regions is very cursory. May I suggest to split this image into four parts (before region 1, region 1, between region 1 and two and region 2)? This would probably also allow to use a somewhat larger scale which

would allow to see finer details (e.g. the "fragile branches" (line 258) Alternatively, the regions may be separated by frames in Fig. 2.

**Reply**: Many thanks for the suggestions. The authors considered different options how to improve the reading and perception of Fig.2 following the reviewer's comment. Eventually, we decided to keep the version with a highlighted blue background for particles associated with the cloud segments 1 and 2.

[Figure]

7.   Line 210ff: Why don´t you show the humidity data in the layer between the cloud layers?

**Reply**: Unfortunately, we do not have a complete RH profile between the cloud layers. During the porpoising the Convair-580 only twice ascended just above the cloud top of the lower layer (i.e., Cc-Ns). These flight segments are outside of the time interval shown in Fig.1.

8.   Line 299: What is the meaning of "480"?

**Reply**: Deleted.

9.   Line 92: Riming might be another process changing the pristine SIP particles.

**Reply**: The sentence was modified following the reviewer's suggestion: "*Depending on their age, the secondary particles experience metamorphoses of shape and size due to varying ambient supersaturation S and temperature T or riming.*"

10.   Line 39 below -38°C

**Reply**: Corrected as per reviewer's comment.

**Response to Reviewer 2 (Alex Khain) of "Observation of secondary ice production in clouds at low temperatures" authored by A. Korolev et al. (egusphere-2022-491)**

The authors thank the reviewer and the editor for the positive evaluation of the paper and their constructive comments. In our response below, the reviewer's comments are in black, and our responses are in blue.

The study provides the first in-situ observation of secondary ice production at temperatures as low as -27°C. These observations are unique and important. I recommend to accept the paper with minor revisions.

Minor comments are:

1.      line 40. The formation of ice by droplet freezing is not mentioned. Since the temperature measured was higher than the temperature of homogeneous drop freezing, the standard immersion drop freezing takes place at T=-27C is. Can the authors evaluate the rate of this immersion freezing of drops (for instance using standard Bigg formula)? Note that according to the observations presented in this study, the zones of high concentration of ice crystals coincide with the zones of significant peaks in droplet concentration and LWC.

Please discuss the possible role of the freezing of drops, which concentration is several orders higher than that of ice crystals, in production of cloud ice. It would be reasonable to refer in this context the study by Khain et al. (2022), in which A. Korolev is a co-author.

**Reply**: The peak of high ice concentration ($N_{ice}$ ) in zone 1 (Fig.1a,b) coincides with the droplet concentration $N_{drop}$ ranging from approximately 10cm$^{-3}$ to 50cm$^{-3}$ (Fig.1c) However, in two other cloud regions (2 and 3) with high $N_{ice}$ (Fig.1a,b), the concentration of droplets $N_{drop}$=0. On the oher hand, there are many cases in Fig.1c, when cloud regions with the droplet concentration peaking to 100cm$^{-3}$ correspond to ice concentration as low as a few particles per L$^{-1}$. Therefore, the authors consider that the reviewer's statement "*according to the observations presented in this study, the zones of high concentration of ice crystals coincide with the zones of significant peaks in droplet concentration and LWC*" is not applicable to the results presented in this study. The absence of liquid water in cloud regions 2 and 3 in Fig.1 makes sense, since in case of high $N_{ice}$, the WBF process will rapidly deplete LWC. For these specific cases, the glaciation time is estimated as 60-90s for the LWC conditions as in the studies cloud layer.

Following the reviewer's comment, the authors performed calculations of the ice production due to droplet freezing based on the Bigg's formula

$$\frac{dN_{ice}(i)}{dt} = aCN_{dr}(i)m_{dr}(i) \exp{(-bT_C)} \qquad \text{(R1)}$$

where $N_{dr}$ and $m_{dr}$ are the concentration and mass of droplets of *i*-th size category, respectively, and $N_{ice}$ is the concentration of ice particles formed due to freezing of the droplets of the *i*-th category, $T_C$ is the air temperature in $^{\text{o}}$C, and $a$=10$^{-4}$ s$^{-1}$g$^{-1}$, b=0.66°C$^{-1}$, $C$=1 are constants.

Integrating Eq.R1 over the droplet size distribution yields:

$$\frac{dN_{ice}}{dt} = \sum_i aCN_{dr}(i)m_{dr}(i)\exp(-bT_C) = aCW\exp(-bT_C) \tag{R2}$$

where W is LWC. Integrated of Eq.R2 over the droplet size distribution shown in Fig.S2a (supplementary material) gives the rate of the droplet freezing $dN_{ice}/dt \approx 0.3$ L$^{-1}$s$^{-1}$. This is a quite high rate of primary ice nucleation. Assuming a residence time of a cloud parcel to be 10min, it yields the concentration of frozen droplets $N_{ice} \approx 180$L$^{-1}$. The obtained value is higher that the 95 percentile of ice concentration (see Supplementary material Fig.S3). However, the obtained ice concentration is significantly lower than N$_{ice}$ in SIP cloud regions 1, 2, and 3 (Fig.1). To address the reviewer's comment, the following text was added:

"Another possibility explaining the enhanced ice concentration may be related to the droplet freezing. The rate of droplet freezing has been assessed here with the help of the Bigg's equation (Bigg, 1953; Khain et al. 2021). For the droplet size distribution averaged over the cloud span (Fig.S2a) it was found that at -27°C the rate of droplet freezing is approximately $dN_{ice}/dt \approx 0.3$ L$^{-1}$s$^{-1}$ (see supplementary material). Therefore, in order to reach an enhance ice concentration of the order of 10$^3$L$^{-1}$ the residence time of the cloud parcel should be $N_{ice}/dN_{ice}/dt \approx 0.92$h. This is an unrealistically long residence time for a cloud parcel in a stratirom cloud layer of a few hundred meter depth. During this time the turbulent diffusion will smear the entire cloud parcel as well as ice particles mitigating formation of sharp gradients of ice concentration as in Fig.1a,b. All these, makes the "droplet freezing" hypothesis insufficient to explain the observed enhanced concentration of ice. "

A text explaining calculations of the rate of droplet freezing was also added in the Supplementary Material.

[Figure]

Can you compare the rates of immersion drop freezing and the rates of the mechanisms of primary ice nucleation mentioned in the paper?

Reply: The immersion drop freezing mechanism refers to the mechanisms of primary ice nucleation. Therefore, the above question is redundant.

However, if the reviewer meant "the mechanism of secondary ice nucleation" then the the assessment of the rates of the SIP mechanisms is hindered due to the absence reliable lab experimental measurements. The review of the status of laboratory studies of SIP were discussed in detail in Korolev and Leisner (2020, ACP). The authors are sceptical about the accuracy of the existing SIP parameterizations.

2.      Line 68. The study Qu et al, (2019) is not presented in the reference list. If you mean the study published in J. Geophys. Res. (see list of references below), that study shows that only secondary ice production can explain in-situ observations. The fraction of ice produced by primary nucleation was evaluated as several per cents at all temperatures. In my view, the study by Qu et al, (2019) was the first paper that reproduced size distributions ice and water observed in-situ measurements and showed that SIP plays a crucial role in the formation of such distributions. So, the statement in the current study that "Such approach may lead to underrepresentation of the role of secondary ice and result in biases in simulations" is not attributed to the study by Qu et al. (2019).

Reply: The missing reference was added:

Qu, Z., Korolev, A., Milbrandt, J. A., Heckman, I., Huang, Y., McFarquhar, G. M., Morrison, H., Wolde, M., and Nguyen C.: The impacts of secondary ice production on microphysics and dynamics in tropical convection. Atmos. Chem. Phys., in press, https://doi.org/10.5194/egusphere-2022-235, 2022

3.      Line 75. Here reference to Qu et al. (2019) as well as to Phillips et al. (2017) should be included. In these studies simulations with a bin-microphysics cloud model reproduced ice

Reply: The references Qu et al. (2019) and Phillips et al.(2017) were added as per reviewer's comment (line 75).

4.      Line 78. The important attempt to understand the fundamental mechanisms of SIP by drop freezing was carried out by Staroselsky et al., 2021.

Reply: The reference Staroselsky et al. was added (line 268)

5.      Line 128. Fig. 1. Please pay attention on the high correlation between droplet concentration and LWC, on the one hand, and the concentration of ice particles. In my opinion, this correlation shows the key role of drops in the formation of ice particle concentration. I believe

that this high correlation decreases the number of possible SIP mechanisms, at least in the present case study.

**Reply**:    Despite occasional peak-to-peak correlations the overall correlation coefficients for the indicated above parameters were found to be quite low. Thus, for the flight segment along the cloud top (excluding SIP regions) the calculated correlation coefficient for two averaging times are shown in the table below.

|  | 1s-averaging | 5s-averaging |
|---|---|---|
| Corr($N_{dropl}$, $N_{ice}$) | 0.070 | 0.079 |
| Corr(LWC, $N_{ice}$) | 0.202 | 0.23 |

Such correlation coefficients are quite low, and they are rather indicative of absence of correlation $N_{dropl}$, LWC and $N_{ice}$. The absence of correlation is suggestive that the involvement of cloud droplets in ice initiation occurs in an essentially non-linear way.

[revised manuscript text omitted]

**Calculation of the rate of droplet freezing.**

Calculation of the rate of droplet freezing performed calculations of the ice production due to droplet freezing based on the Bigg's equation

550 $\qquad \frac{dN_{ice}(i)}{dt} = aCN_{dr}(i)m_{dr}(i)\exp(-bT_C)$ $\qquad\qquad\qquad\qquad\qquad\qquad\qquad$ (S1)

where $N_{dr}$ and $m_{dr}$ are the concentration and mass of droplets of *i*-th size category, respectively, and $N_{ice}$ is the concentration of ice particles formed due to freezing of the droplets of the *i*-th category, $T_C$ is the air temperature in °C, and $a$=10$^{-4}$ s$^{-1}$g$^{-1}$, b=0.66°C$^{-1}$, $C$=1 are constants.

Integrating Eq.S1 over the droplet size distribution yields:

555 $\qquad \frac{dN_{ice}}{dt} = \sum_i aCN_{dr}(i)m_{dr}(i)\exp(-bT_C) = aCW\exp(-bT_C)$ $\qquad\qquad\qquad$ (S2)

where W is LWC.

Integrated of Eq.S2 over the droplet size distribution shown in Fig.S2a gives the rate of the droplet freezing $dN_{ice}/dt \approx 0.3$ L$^{-1}$s$^{-1}$.

[Figure]

[Figure]

560 Figure S4. (a) Droplet size distribution (same as in Fig.S2a). (b) Distribution of the rates of droplet freezing corresponding to the droplet size distribution in (a) and $T_c$=-27°C.